# Model-Based Design Approach to Improve Performance Characteristics of Hydrostatic Bearing Using Multivariable Optimization

**Waheed Ur Rehman** [1,2,*], **Xinhua Wang** [1,2,*], **Yiqi Cheng** [1,2], **Yingchun Chen** [1,2], **Hasan Shahzad** [1,2], **Hui Chai** [1,2], **Kamil Abbas** [1,2], **Zia Ullah** [1,2] **and Marya Kanwal** [1,2]

1   College of Mechanical Engineering and Applied Electronics Technologies, Beijing University of Technology, Beijing 100124, China; chengyiqi1993@163.com (Y.C.); ychen08089@163.com (Y.C.); Hasanshahzad99@hotmail.com (H.S.); henry0806@126.com (H.C.); kamil_abbas3@hotmail.com (K.A.); engineerziaullah@yahoo.com (Z.U.); maryakanwal55@hotmail.com (M.K.)
2   Faculty of Materials and Manufacturing, Beijing University of Technology, Beijing 100124, China
*   Correspondence: wrehman87@bjut.edu.cn (W.U.R.); wangxinhua@bjut.edu.cn (X.W.)

**Abstract:** Research in the field of tribo-mechatronics has been gaining popularity in recent decades. The objective of the current research is to improve static/dynamics characteristics of hydrostatic bearings. Hydrostatic bearings always work in harsh environmental conditions that effect their performance, and which may even result in their failure. The current research proposes a mathematical model-based system for hydrostatic bearings that helps to improve its static/dynamic characteristics under varying conditions of performance-influencing variables such as temperature, spindle speed, external load, and clearance gap. To achieve these objectives, the capillary restrictors are replaced with servo valves, and a mathematical model is developed along with robust control design systems. The control system consists of feedforward and feedback control techniques that have not been applied before for hydrostatic bearings in the published literature. The feedforward control tries to remove a disturbance before it enters the system while feedback control achieves the objective of disturbance rejection and improves steady-state characteristics. The feedforward control is a trajectory-based controller and the feedback controller is a sliding mode controller with a PID sliding surface. The particle swarm optimization algorithm is used to tune the 6-dimensional vector of the tuning parameters with multi-objective performance criteria. Numerical investigations have been carried out to check the performance of the proposed system under varying conditions of viscosity, clearance gap, external load and the spindle speed. The comparison of our results with the published literature shows the effectiveness of the proposed system.

**Keywords:** multivariable optimization; tribo-mechatronics; hydrostatic bearing; active lubrication; sliding mode control; fluid film thickness

## 1. Introduction

The use of mechatronics to improve tribological performance of hydrostatic bearings is called tribo-mechatronics. Research in the field of tribo-mechatronics has been gaining popularity in recent decades [1]. Once a mechanical system has been designed, then it is hard to improve the static/dynamics characteristics. As such, scientists are looking for mechatronics methods to improve performance of fixed mechanical systems, such as a traditional hydrostatic journal bearing. Enhancing the performance of the hydrostatic journal bearings using different methodologies is a hot topic.

Several authors have tried to improve the static/dynamic characteristics of hydrostatic bearings using different techniques and methods, and an overview of their achievement can be found in this paragraph. The dynamic characteristics are improved with the help of variable compensation, which is provided by using a double-action restrictor of the tapered

spool type and the cylindrical spool type [2,3] but the deficiency of this approach is the slow response time of the spool type restrictor as compared to the servo valve. A numerical study is carried out to find the appropriate value of the recess shape, the strength of the magnetic field and the restrictor for the optimal performance of hydrostatic bearings [4]. A shaft's eccentricity is controlled with the help of a piezoelectric membrane restrictor [5]. A series of studies have been undertaken on active hybrid bearings, in which the part one describes the mathematical model and second part outlines the control design techniques for improving dynamic characteristics [6,7]. The performance characteristics of the hydrostatic bearings are checked against different shapes of recess [8,9] but there is a performance limitation due to the fixed throttling effect. A prototype is developed for the active bearing where the clearance gap is controlled through high-speed solenoid valves but deficiency arises because it is used for light load applications [10]. The rotor bearing node's characteristics are improved by the integration of control techniques for thrust fluid bearings [11]. The eddy current sensors are used to monitor the gap of the hydrostatic bearing so that the load-carrying capacity can be improved by activating the hydrostatic support but there is deficiency in the low resolution of the eddy current sensors [12]. A servo control feedback system is introduced to monitor the fluid film gap and advance control techniques are proposed to improve the dynamic characteristics [13,14], but the deficiency is the lack of experimental validation. Magnetic fluid is used as an active lubricant input to control the fluid film thickness for achieving the high load-carrying capacity, but the drawback is that it only works with magnetic fluids [15]. Nonlinear mathematical models are proposed, and a comparison analysis is performed under different types of control inputs for the active hydrostatic thrust bearing [16]. The performance of the opposed pad hydrostatic bearing is improved by using a membrane-type hydrostatic restrictor [17]. The $H_\infty$ loop and inner loop methods are used for stabilizing the shaft and reducing vibrations [18,19]. Integration of tribo-mechatronics and control has divided the active hydrostatic bearings into three classes. In class one, researchers have tried to use the external magnetic field for controlling viscosity of the magnetic fluid, so that the shaft vibrations could be controlled [20–24]. In the second class, authors tried to control recess pressure, and efforts were made to control the recess pressure through a piezoelectrically activated jet [25,26]. An active inherent restrictor was incorporated into the body of the bearing to achieve purpose of high stiffness and good rotational accuracy of the bearing [27]. In the third class, authors have tried to change the external supply pressure with the help of hydraulic actuator as an active device so that stiffness and the load-carrying capacity could be increased [28–30].

The current research presents a model-based design approach whereby a mathematical model is created, and then performance characteristics are improved with the help of advanced control techniques and optimizing algorithms. The effectiveness is verified by comparing results with those of the previous published literature. This paper's contents are organized as follows. Section 2 outlines the design of a mathematical model. Section 3 presents control techniques including the multivariable performance criteria and optimizing algorithm. Section 4 presents the results and discussion. The latter part of the paper is followed by conclusions, acknowledgement, and references.

## 2. Mathematical Model

The details of the working principle for active hydrostatic bearing can be found in existing literature [31]. The hydrostatic journal bearing with a servo control mechanism is shown in Figure 1. The flow chart on which Figure 1 is based is shown in Figure 2, where two servo valves are responsible for the horizontal bearing clearance and two valves are responsible for the vertical bearing clearance. The external load is also divided into horizontal and vertical components. The controllers are tuned with the help of a particle swarm optimization algorithm. The displacement sensors measure the bearing clearance with the help of shaft displacement. The error signal is generated by the comparison of the reference signal and the signal of displacement sensor. The controllers use error signal to

drive spool of the servo valve, so that the desired fluid flow can be delivered to hydrostatic bearing to achieve the required bearing clearance.

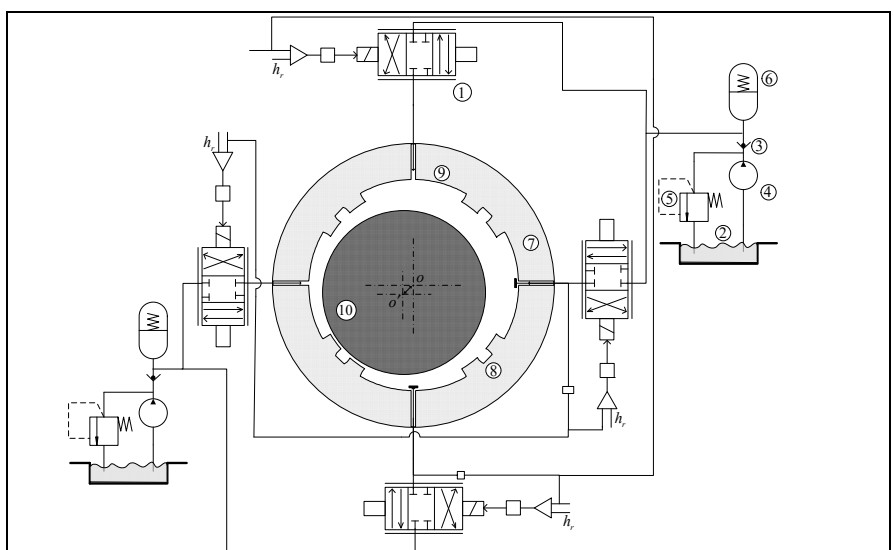

**Figure 1.** Structure of hydrostatic bearing ① Servo Valve ② Oil Tank ③ Check Valve ④ Pump ⑤ Over Flow valve ⑥ Accumulator ⑦ Journal Bearing ⑧ Return Chute ⑨ Recess ⑩ Shaft.

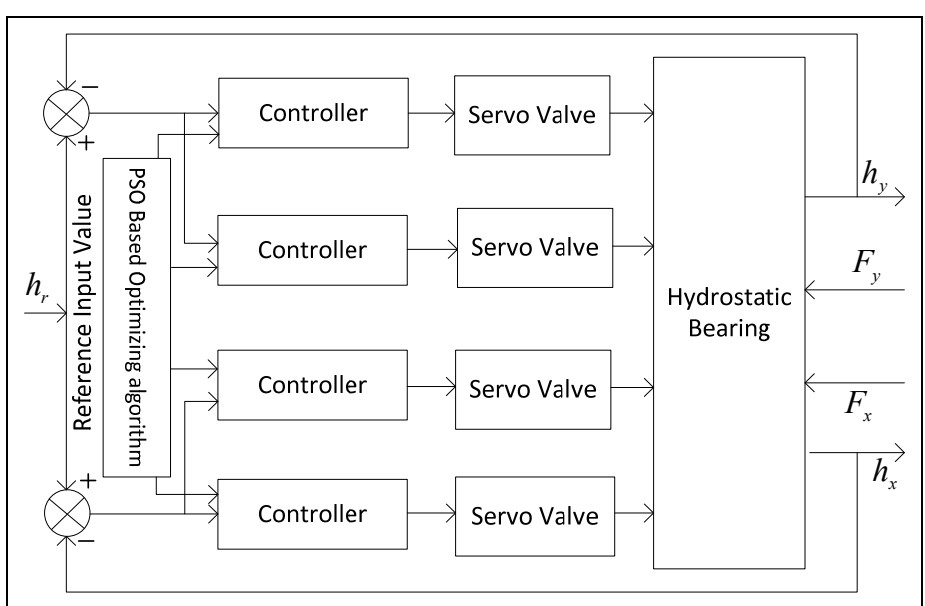

**Figure 2.** Flow Chart for proposed system.

First, the mathematical model of the electrohydraulic flow control servo valve is derived. The second-order mathematical model of the servo valve may be described as follows [32,33]:

$$\ddot{v}_i + 2\varsigma_v \omega_v \dot{v}_i + \omega_v^2 v_i = \omega_v^2 k_v u_i \qquad \therefore i \in x, y \tag{1}$$

where $u_i$ is control input to the servo valve, $\zeta_v$ is the damping factor, $k_v$ is the proportional constant, $\omega_v$ is natural frequency and $v_i$ is spool displacement. The motion of the spool of the servo valve is proportional to flow when external load is constant. The flow of fluid is

proportional to the pressure drop prior to the condition of varying load. The relationship between the input, control flow, and pressure drop is given by the equation

$$
\begin{cases}
Q_{v2} = c_d w v_y [2(P_s - P_2)/\rho]^{1/2} \\
Q_{v4} = c_d w v_y [2(P_s - P_4)/\rho]^{1/2} \\
Q_{v1} = c_d w v_x [2(P_s - P_1)/\rho]^{1/2} \\
Q_{v3} = c_d w v_x [2(P_s - P_3)/\rho]^{1/2}
\end{cases}
\tag{2}
$$

Applying the fluid lubrication theory to get the flow across axial and circumferential land,

$$
Q_{i-out} = \frac{h_i^3 P_i}{6\eta}\left(\frac{l}{b_1} + \frac{b}{l_1}\right) + \frac{u}{2} l h_i
\tag{3}
$$

One can use the continuity equation to describe overall flow which includes compressible and squeezing flow:

$$
c_d w x_{vi} \sqrt{\frac{2}{\rho}(P_s - P_i)} = Q_{i-out} + A_e \frac{dh_i}{dt} + \frac{V_e}{\beta_e}\frac{dp_i}{dt}
\tag{4}
$$

Substituting Equation (3) into Equation (4);

$$
c_d w x_{vi} \sqrt{\frac{2}{\rho}(P_s - P_i)} = \frac{h_i^3 P_i}{6\eta}\left(\frac{l}{b_1} + \frac{b}{l_1}\right) + \frac{u}{2} l h_i + A_e \frac{dh_i}{dt} + \frac{V_e}{\beta_e}\frac{dp_i}{dt} \qquad \therefore i \in 1,2,3,4
\tag{5}
$$

The Reynold equation for the hydrostatic bearing can be described as follows:

$$
\frac{\partial}{\partial x}\left(\frac{h^3}{\eta}\frac{\partial p}{\partial x}\right) + \frac{\partial}{\partial z}\left(\frac{h^3}{\eta}\frac{\partial p}{\partial z}\right) = 6v\frac{\partial h}{\partial x} + 12\dot{h}
\tag{6}
$$

The total pressure is a combination of oil chamber pressure, axial pressure, circumferential chamber and cavity thrust, and is given by

$$
p_i(t) = p_{cavity} + p_{circumferential} + p_{axial}
\tag{7}
$$

Axial pressure, cavity pressure and circumferential pressure are given by [31]

$$
p_{axial} = 2(b + b_1)\left[-l_1^3 \frac{\eta \dot{h}_i}{h_i^3} + p_i(l + \frac{3}{2}l_1)\right]
\tag{8}
$$

$$
p_{circumferential} = 2(l + l_1)\left[-b_1^3 \frac{\eta \dot{h}_i}{h_i^3} + p_i(b + \frac{3}{2}b_1)\right]
\tag{9}
$$

$$
p_{cavity} = l b p_i
\tag{10}
$$

Substituting Equation (8) to Equation (10) in Equation (7) and equating $D_e = -\frac{2\eta}{h_0^3}\left[b_1^3(l + l_1) + l_1^3(b + b_1)\right]$;

$$
P_i(t) = A_e p_i + D_e \dot{h}_i \qquad \therefore i \in 1,2,3,4
\tag{11}
$$

The expansion region has negative dynamic pressure with positive displacement and vice versa. One can describe it as follows:

$$
\begin{cases}
\Delta p_1 = -\Delta p_3 = -\Delta p_y \\
\Delta p_2 = -\Delta p_4 = -\Delta p_x
\end{cases}
\tag{12}
$$

Suppose that the external load which produces eccentricity is $e$ and it acts with an angle $a$. The change in fluid film thickness due to external load is given by

$$\Delta h_1 = e \cos a, \quad \ddot{e} = \frac{\ddot{h}_1}{\cos \alpha} \tag{13}$$

Substituting Equations (12) and (13) into Equation (5);

$$\begin{cases} c_d w x_{vn} \sqrt{\frac{2}{\rho}(P_s - P_0 \pm p_j)} = Q_{n-out} \pm A_e \dot{e} \cos a \mp \frac{V_e}{\beta_e} \dot{p}_j \\ c_d w x_{vm} \sqrt{\frac{2}{\rho}(P_s - P_0 \pm p_k)} = Q_{m-out} \pm A_e \dot{e} \sin a \mp \frac{V_e}{\beta_e} \dot{p}_k \end{cases} \quad \therefore \quad \begin{matrix} n \in 1,3 \\ j \in y \\ m \in 2,4 \\ k \in x \end{matrix} \tag{14}$$

By applying binomial theorem expansion to Equation (14);

$$\begin{cases} c_d w x_{vn} \sqrt{\frac{2}{\rho}(P_s - P_0)}\left(1 \pm \frac{p_j}{2(P_s - P_0)}\right) = Q_{n-out} \pm A_e \dot{e} \cos a \mp \frac{V_e}{\beta_e} \dot{p}_j \\ c_d w x_{vm} \sqrt{\frac{2}{\rho}(P_s - P_0)}\left(1 \pm \frac{p_k}{2(P_s - P_0)}\right) = Q_{m-out} \pm A_e \dot{e} \sin a \mp \frac{V_e}{\beta_e} \dot{p}_k \end{cases} \quad \therefore \quad \begin{matrix} n \in 1,3 \\ j \in y \\ m \in 2,4 \\ k \in x \end{matrix} \tag{15}$$

Substituting Equation (12) into Equation (11);

$$\begin{cases} \Delta p_n(t) = \mp A_e \Delta p_j \pm D_e \dot{e} \cos a \quad \therefore n \in 1,3 \,\&\, j \in y \\ \Delta p_m(t) = \mp A_e \Delta p_k \pm D_e \dot{e} \sin a \quad \therefore m \in 2,4 \,\&\, k \in x \end{cases} \tag{16}$$

Motion dynamics of spindle of hydrostatic bearing when the external load acts on the shaft is given by [24]:

$$\left[F(t) + m \frac{\ddot{h}_1}{\cos \alpha}\right] \cos \alpha - D_e(\dot{h}_3 - \dot{h}_1) = 2 A_e \Delta p_y \tag{17}$$

## 3. Proposed Control Strategy

The proposed control scheme consists of two types of control inputs, sliding mode control and feedforward control. The proposed control scheme is given by;

$$u = u_{smc} + u_f \tag{18}$$

where $u_{smc}$ is sliding mode control and $u_f$ is feed forward control.

### 3.1. Sliding Mode Control

Sliding mode control is a technique in which two phases appear, such as a sliding phase and a reaching phase. Sliding phase has $s(t) = 0$. The reaching face appears when the system moves towards the sliding phase. The efficiency of sliding mode control depends on the sliding surface. The current system will use a PID type sliding surface which will be given by

$$s(t) = k_p e(t) + k_i \int_0^t e(t) dt + k_d \dot{e}(t) \tag{19}$$

One can apply sign function to the sliding surface to get sliding mode control [34]:

$$u_{smc} = \lambda s(t) + k_s \text{sgn}(\dot{s}(t)) \quad \lambda, k_s \in \Re^+ \tag{20}$$

where $\text{sign}(\dot{s}(t))$ is piecewise function which is given by:

$$sign(\dot{s}(t)) = \begin{cases} 1 & ; \dot{s}(t) > 0 \\ 0 & ; \dot{s}(t) = 0 \\ -1 & ; \dot{s}(t) < 0 \end{cases} \tag{21}$$

Let us suppose there is a Lyapunov function such as

$$V = \frac{1}{2}s^2 \tag{22}$$

Taking derivative;

$$\dot{V} = s\dot{s} \tag{23}$$

By putting Equation (19) into (23), it is shown that one must choose the value of $k_p, k_i$ and $k_d$ which will converge the sliding surface value to the zero. When the sliding surface value approaches zero then it will show the sliding surface phase rather than reaching phase. The suppression of chattering increases the stability, and one can reduce the chattering effect in Equation (20) by introducing a hyperbolic tangent function with ɸ as boundary layer [34].

$$u_{smc} = \lambda s(t) + k_s \tanh\left(\frac{\dot{s}(t)}{\varphi}\right) \quad \lambda, k_s \in \Re^+ \tag{24}$$

### 3.2. Feedforward Control

The current system has a disturbance in the form of external load which creates eccentricity. Feedforward control is used to improve the disturbance rejection performance of proposed system. The feedforward controller can effectively decrease the impact of disturbance as compared to feedback control [35]. The feedforward controller has an advantage that is to remove disturbance before entering systems while in a feedback controller disturbances travel through the whole system and are rejected by error of feedback signal [36]. The current system derives a trajectory-based feedforward controller, which uses the dynamics of the mathematical model of the proposed system.

$$
\begin{aligned}
u_{f1} &= \frac{-\frac{V_b}{\beta}\Delta\dot{p}_{y1}}{\sqrt{\frac{2}{\rho}}(P_s-P_0)\left(\frac{C_d wk_{sv}\omega^2}{\omega^2+2\varsigma\omega+1}\right)\left(1+\frac{1}{2}\frac{\Delta p_y}{(P_s-P_0)}\right)} \\
u_{f3} &= \frac{\frac{V_b}{\beta}\Delta\dot{p}_y}{\sqrt{\frac{2}{\rho}}(P_s-P_0)\left(\frac{C_d wk_{sv}\omega^2}{\omega^2+2\varsigma\omega+1}\right)\left(1-\frac{1}{2}\frac{\Delta p_y}{(P_s-P_0)}\right)}
\end{aligned}
\tag{25}
$$

### 3.3. Performance Criteria

There are different types of performance criteria which are used to design a PID-based sliding mode controller. Some famous performance criteria are integral of square error (ISE), integral of the time weighted square error (ITSE), and integral of absolute error (IAE). The current research will adopt a multiobjective performance criterion which consists of several performance variables such as setting time, rise time, overshoot and tracking error. These performance variables are measured online during simulation and updated after each simulation to find the value of the objective function, which is given by

$$f(k) = w_1\left[\sum_0^t (h_r-h_x)^2 + \sum_0^t (h_r-h_y)^2\right] + w_2\left[|T_{sx}|+|T_{sy}|\right] + w_3\left[|T_{rx}|+|T_{ry}|\right] + w_4\left[|O_{sx}|+|O_{sy}|\right] \quad \therefore k \in [k_p, k_i, k_d, \lambda, \varphi, k_s] \tag{26}$$

### 3.4. Particle Swarm Optimization

The tuning of parameters of controller with the help of swarm optimization is gaining popularity in the field of automation applications [37,38]. In order to tune the parameters of PID surface-based sliding mode controllers, a six-dimensional vector is introduced for $k_p$, $k_i$, $k_d$, $\lambda$, $\varphi$, $k_s$. This six-dimensional vector is optimized with respect to objective function using particle swarm optimization. Here are some terms which are important to define for particle swarm optimization.

① $\Theta = [\theta_1, \theta_2, \theta_3, \theta_4, \theta_5, \theta_6] = [k_p, k_i, k_d, \lambda, \varphi, k_s]$. This is most basic element of PSO algorithm. It represents a solution for tuning of parameters for PID surface-based

sliding mode controllers. At the nth iteration, the ith particle is denoted by $\Theta_i(n) = [\theta_{i,1}(n), \theta_{i,2}(n), \theta_{i,3}(n), \theta_{i,4}(n), \theta_{i,5}(n), \theta_{i,6}(n)]$. The optimizing parameters are bounded by upper and lower bound $[\theta_{\min}, \theta_{\max}]$ during simulation of proposed system.

$$\theta_j = \begin{cases} \theta_{\min} & if \ \theta_j < \theta_{\min} \\ \theta_j & if \ \theta_{\min} \leq \theta_j \leq \theta_{\max} \\ \theta_{\max} & if \ \theta_j > \theta_{\max} \end{cases} \tag{27}$$

② Velocity $V(n)$: it shows the moving velocity of particle $\Theta(n)$. At the $n$th iteration, the $i$th particle velocity is given by

$$V_i(n) = [v_{i,1}(n), v_{i,2}(n), v_{i,3}(n), v_{i,4}(n), v_{i,5}(n), v_{i,6}(n)] \tag{28}$$

③ Individual best $P(n)$: the particle compares the cost function with the best one when it moves in space. The particle with the best cost function is called individual best. The individual best for the $i$th particle is solved in such way that it holds

$$J(P_i(n)) \leq J(\Theta_i(\tau)) \quad \therefore \tau \leq n \tag{29}$$

where $J(P_i)$ and $J(\Theta_i)$ are cost function for $P_i$ and $\Theta_i$. At the nth iteration, the ith individual best is given by

$$P_i(n) = [p_{i,1}(n), p_{i,2}(n), p_{i,3}(n), p_{i,4}(n), p_{i,5}(n), p_{i,6}(n)] \tag{30}$$

④ Global best $G(n)$: the global best value is the best value among all individual best values. At the $n$th iteration, the global best $G(n) = [g_1(n), g_2(n), g_3(n), g_4(n), g_5(n), g_6(n)]$ is solved in such a way that

$$J(G(n)) \leq J(P_i(n)) \quad \therefore i = 1, 2, , H \tag{31}$$

⑤ Velocity and position of the particle are updated according to individual best and global best position. The velocity and position of the ith particle are given by

$$\begin{aligned} v_{i,j}(n+1) &= wv_{i,j}(n) + c_1 r_1 \left(p_{i,j}(n) - \theta_{i,j}(n)\right) + c_2 r_2 \left(g_j(n) - \theta_{i,j}(n)\right) && \therefore i = 1, 2, 3 \dots, H \\ \theta_{i,j}(n+1) &= \theta_{i,j}(n) + v_{i,j}(n+1) && \therefore j = 1, 2, 3, \end{aligned} \tag{32}$$

where $v_{i,j}(n)$ is a current velocity, $v_{i,j}(n+1)$ is the next velocity, $w$ is inertial weight, $c_1$ and $c_2$ are positive acceleration constants, $r_1$ and $r_2$ are two random numbers.

⑥ Termination condition: There are two termination conditions. The first one is to achieve the required value of objective function while the second one is to achieve the required number of iterations. The current research will prefer to achieve require number of iterations so that the given objective function can be reduce to minimum cost value.

The whole design steps to tune parameters for PID sliding surface-based sliding mode controller are as follows.

Step 1: Define an objective function. The objective function is calculated by performing a simulation online in Simulink and updating monitoring variables such as rise time, settling time, overshoot and error to the MATLAB workspace. The parameters of PSO such as $w, c_1, c_2$, and number of iterations are defined.

Step 2: If the pre-defined number of iterations have been achieved, then stop the PSO algorithm; else, update objective function by performing simulation and update monitoring variables such as rise time, settling time, overshoot and error to the MATLAB workspace.

Step 3: Find the value of individual best such that it holds the condition of Equation (29).

Step 4: Find the value of global best such that it holds the condition of Equation (31).

Step 5: Update velocity and position of the particles using Equation (32) and apply upper and lower bound to Equation (27) if it goes beyond the interval.

Step 6: Go to step 2.

## 4. Results and Discussion

The rotating machinery normally works under different types of working conditions such as different rotational speed, changing temperature, varying initial oil pressure and external load. So, performance of proposed hydrostatic bearing is checked by using simulation parameters which are given by Yang et al. [31]. The experimental results for traditional hydrostatic journal bearing and hydrostatic journal bearing with PID control are taken from literature [31].

### 4.1. Influence of Varying External Load

There are certain factors which stop hydrostatic bearing from attaining a new equilibrium position after application of the external load. These factors are throttling effect, the mass of the components of the bearing, inertial effects, and hydrostatic stiffness. Mostly efficiency of hydrostatic bearing is measured in term eccentricity and response time to achieve a new equilibrium position when external load is applied. The efficiency of hydrostatic bearing with PID surface-based sliding mode control (SMC-PID) is checked in MATLAB/Simulink by using external load of 800 N and 1600 N, as shown in Figures 3 and 4. The simulation results show that when external load is applied, then active hydrostatic bearing rapidly obtains new equilibrium with zero eccentricity while traditional hydrostatic bearing has a small amount of eccentricity when adjusted to new equilibrium position. The maximum amount of eccentricity for SMC-PID, PID and traditional bearing is $e_1 = 0.50$ um, $e_2 = 0.88$ um and $e_3 = 1.15$ um under 800 N load. The amount of eccentricity for SMC-PID, PID and traditional bearing is $e_1 = 0.87$ um, $e_2 = 1.82$ um and $e_3 = 2.80$ um under 1600 N load. The results under different external load show that active hydrostatic bearing under SMC-PID has less eccentricity as compared to traditional hydrostatic bearing and active hydrostatic journal bearing under PID. The efficiency is measured in terms of "response time for dynamic equilibrium" and "eccentricity". The response time is the time which is required to reduce vibrations produced in fluid film due to external loads. The response time to settle vibrations and achieve dynamic equilibrium is $T_1 = 200$ s and $T_2 = 300$ s for SMC-PID and PID. It shows that active hydrostatic journal bearing under SMC-PID has a faster response to achieve the dynamic equilibrium position.

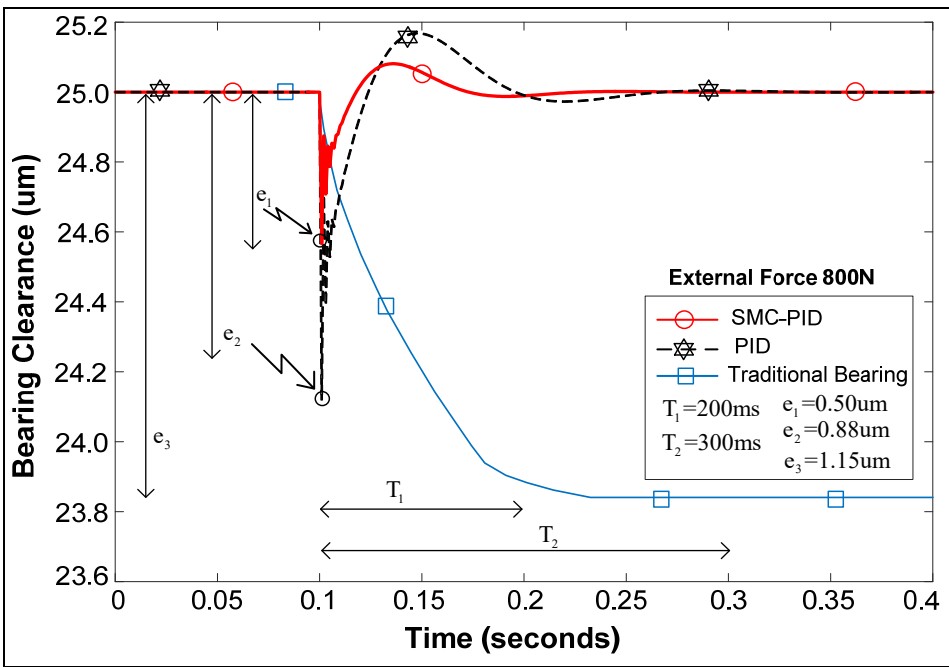

**Figure 3.** Load rejection performance under 800 N.

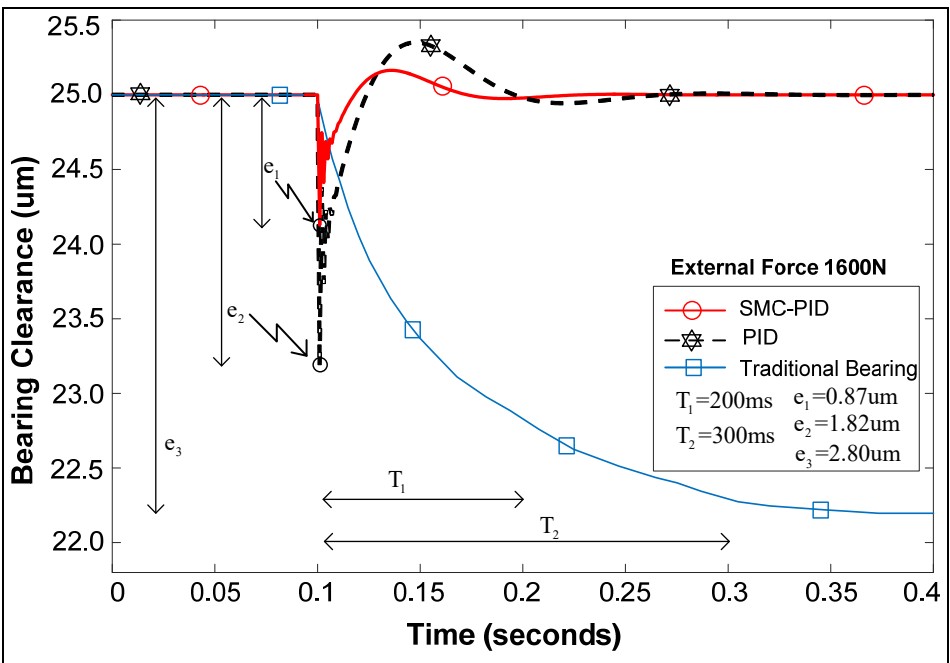

**Figure 4.** Load rejection performance under 1600 N.

## 4.2. Influence of Viscosity

Whenever hydrostatic journal bearing starts its operation, then a pressurized film of oil helps to keep separate the outer surface of the spindle and the inner surface of bearing. The hydrostatic bearing characteristics such as stiffness and damping depend upon this pressurized oil film. This oil film changes its viscosity due to temperature. When hydrostatic bearing runs at high speed, then heat is produced which changes the viscosity of fluid. The simulations are performed for active hydrostatic bearing under different viscosity and results were compared with experimental results of traditional hydrostatic bearing, as shown in Figures 5 and 6.

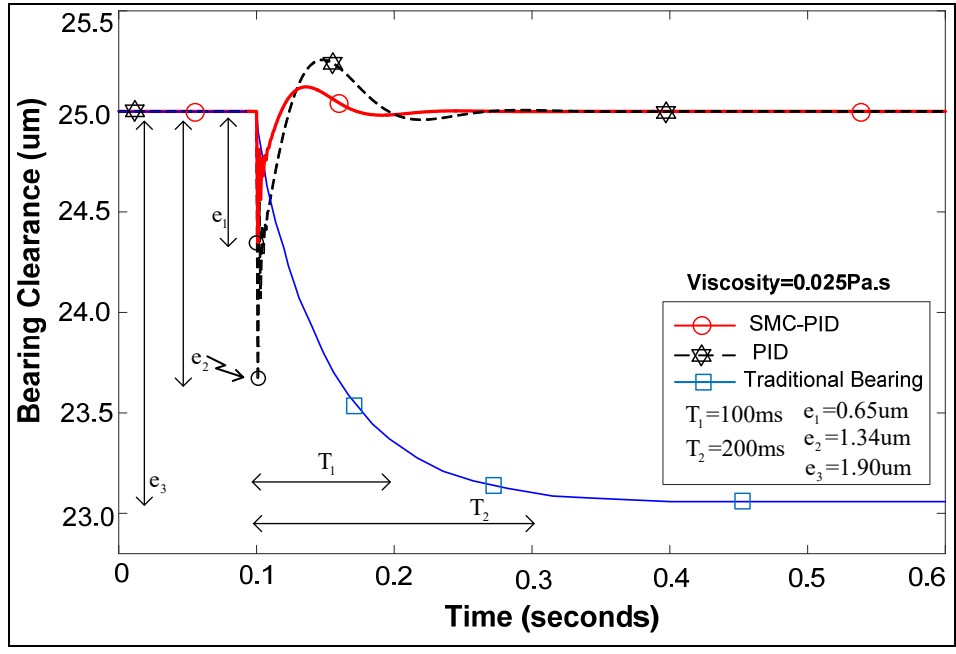

**Figure 5.** Hydrostatic bearing performance under 0.025 Pas Viscosity.

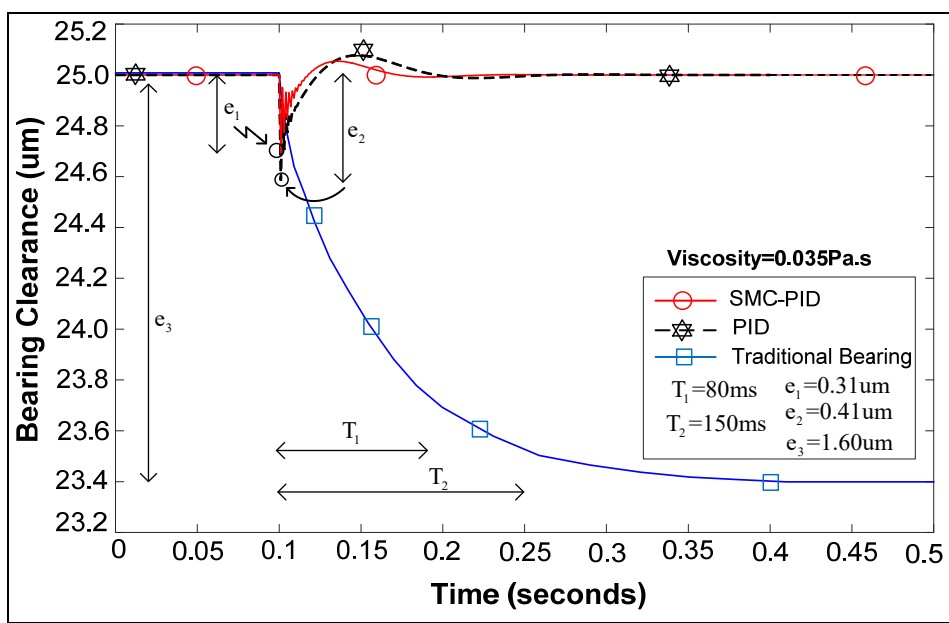

**Figure 6.** Hydrostatic bearing performance under 0.035 Pas Viscosity.

The maximum amount of eccentricity for SMC-PID, PID and traditional bearing is $e_1 = 0.65$ um, $e_2 = 1.34$ um and $e_3 = 1.90$ um under 0.025 Pas while this amount changes to $e_1 = 0.31$ um, $e_2 = 0.41$ um and $e_3 = 1.60$ um under 0.035 Pas. This clearly shows that when the viscosity increases due to the temperature, then there is a decrease in the eccentricity of the hydrostatic bearing. It shows that the performance of the hydrostatic bearing is improved with the smaller increment in viscosity value due to the temperature. The results also show that under same value of viscosity, the performance of hydrostatic bearing is better under SMC-PID as compared to PID and traditional bearing. The response time that is $T_1 = 100$ s and $T_1 = 80$ s for SMC-PID under 0.025 Pas and 0.035 Pas. The response time shows that active hydrostatic bearing under SMC-PID technique quickly suppress the vibrations and achieves dynamic equilibrium with a shorter response time.

### 4.3. Influence of Varying Clearance Gap

The harsh working conditions, such as high speed, heating effect and friction, normally cause an increase in the bearing clearance with the passage of time. The traditional or conventional hydrostatic bearing depends on the oil pressure and the oil film thickness while active hydrostatic bearing depends on the opening of the servo valve, oil film thickness and oil pressure. When the thickness of the oil film (bearing clearance) increases, then the active hydrostatic bearing adjusts the opening of the servo valve in such a way that the stiffness can be improved. The simulation experiments are performed to check the performance of active hydrostatic bearing under SMC-PID control. The maximum amount of eccentricity for SMC-PID, PID and traditional bearing is $e_1 = 0.65$ um, $e_2 = 1.34$ um and $e_3 = 2.00$ um under 25 um bearing clearance, respectively. The amount of eccentricity for SMC-PID, PID and traditional bearing is $e_1 = 0.55$ um, $e_2 = 1.30$ um and $e_3 = 1.93$ um under 25.2 um bearing clearance, respectively. A comparison of Figures 7 and 8 shows that a small increment in bearing clearance shows greater improvement in bearing stiffness for active hydrostatic bearing under SMC-PID. It can be seen that the improvement in reduction in eccentricity is 0.1 um, 0.04 um, 0.07 um for SMC-PID, PID and traditional bearing with an increment of 0.2 um bearing clearance, respectively. The amount of reduction in eccentricity shows that hydrostatic bearing under SMC-PID has more stiffness when bearing clearance increases from 25 um to 25.2 um.

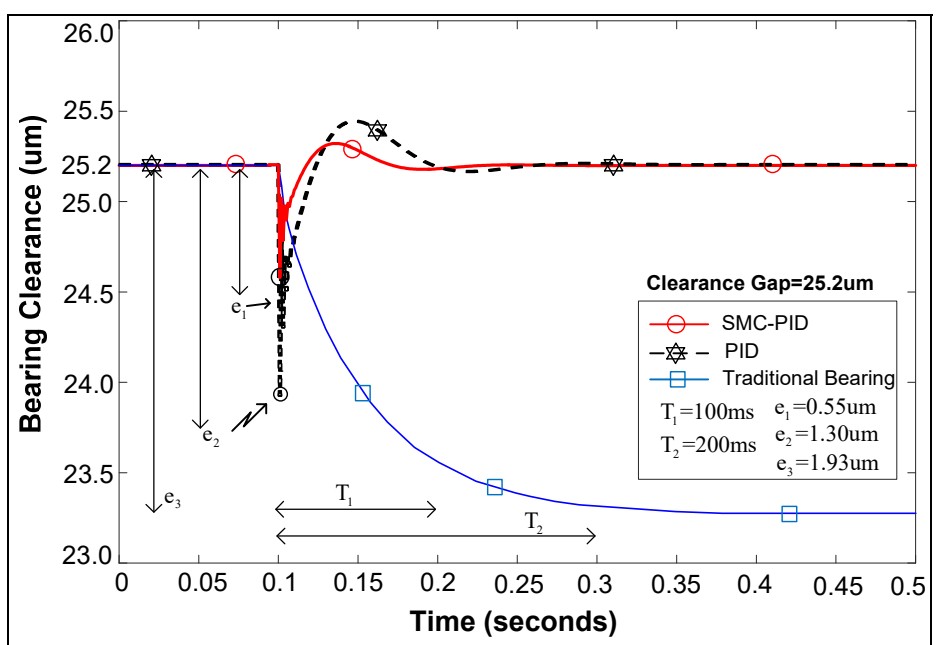

**Figure 7.** Hydrostatic bearing performance under 25.2 um bearing clearance.

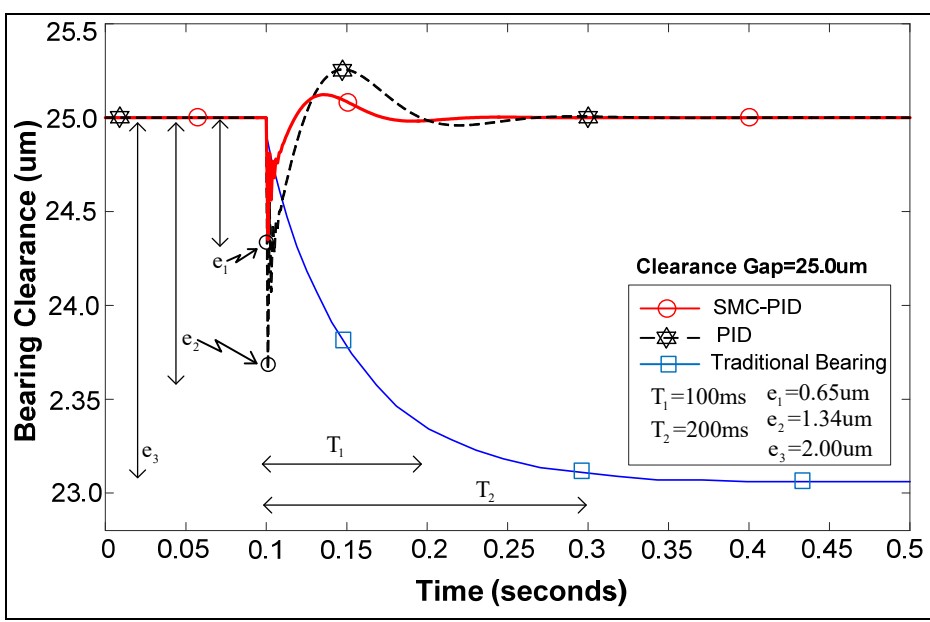

**Figure 8.** Hydrostatic bearing performance under 25 um bearing clearance.

### 4.4. Influence of Variable Spindle Speed

Whenever hydrostatic bearing runs at high speed in the presence of eccentricity, then squeezing effects come into play, and it increase oscillation amplitude of vibrations which are produced due to eccentricity. In conventional or traditional hydrostatic bearing, a fixed throttling device is used. So, that is why the hydrodynamic effect is prominent while in active hydrostatic bearing active throttling device (servo valve), the hydrodynamic effect is dominant. To check the efficiency of hydrostatic bearing under the different speeds, simulation experiments were performed and the results are shown in Figures 9 and 10. The results show that the maximum amount of the eccentricity for SMC-PID, PID and traditional bearing is $e_1 = 0.66$ um, $e_2 = 1.35$ um and $e_3 = 2.57$ um under 800 RPM while this amount changes to $e_1 = 0.65$ um, $e_2 = 1.34$ um and $e_3 = 1.88$ um under 1600 RPM. This clearly shows that when the spindle speed increases, then there is a decrease in the

eccentricity of hydrostatic bearing due to the hydrodynamic effect and the squeezing effect. The results show that the amount of eccentricity is smaller for SMC-PID as compared to PID and traditional bearing. The response time is not affected much more by the speed of the spindle. However, a slight increase in the vibration amplitude is observed which is due to resonance phenomenon, and it gets prominent at higher speeds. The response shows that the active hydrostatic bearing under SMC-PID technique quickly suppresses the vibrations and achieves the dynamic equilibrium with a shorter response time.

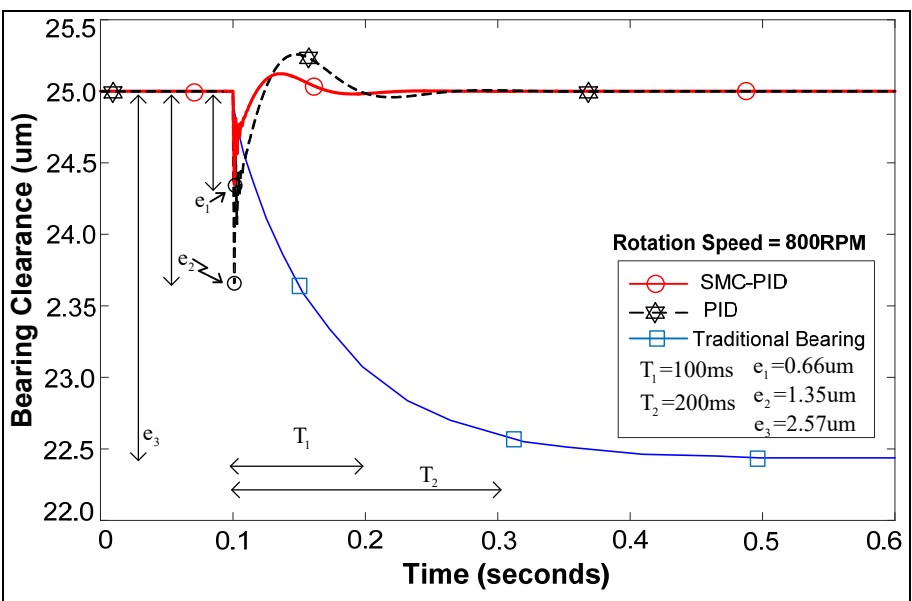

**Figure 9.** Hydrostatic bearing performance under 800 RPM.

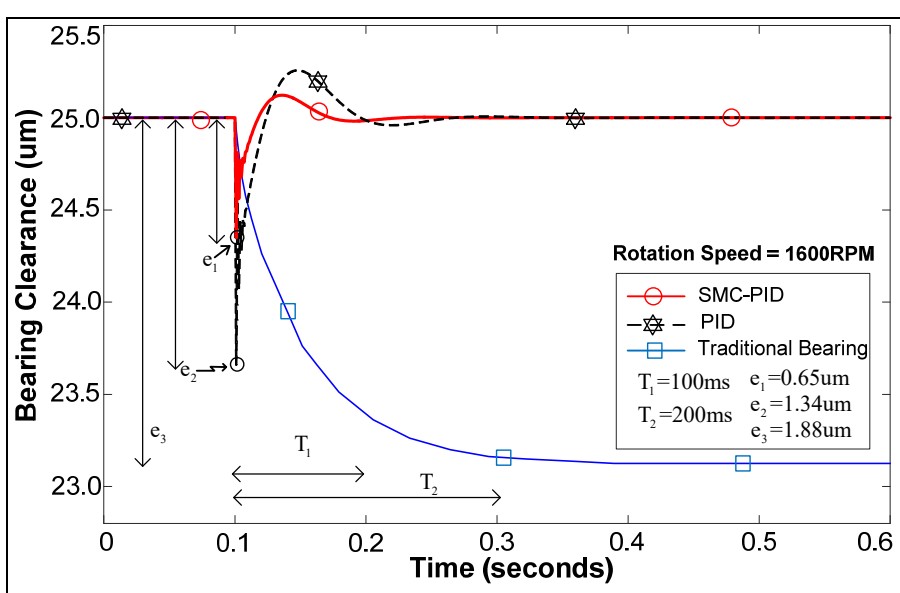

**Figure 10.** Hydrostatic bearing performance under 1600 RPM.

### 4.5. Load Carrying Capacity

Whenever it comes to the efficiency of the hydrostatic bearing, then the load-carrying capacity plays an important role. In order to check the load-carrying capacity of the proposed active hydrostatic bearing under SMC-PID controller, a number of different loads are applied, and the eccentricity ratio ($\varepsilon$) is measured. A graph is plotted between the load parameter ($\overline{w}$) and the eccentricity ratio ($\varepsilon$), as shown in Figure 11. The results of

active hydrostatic journal bearing are compared with the results of capillary-controlled hydrostatic bearing from published literature [39,40]. The comparison of results shows that the active hydrostatic bearing has higher load-carrying capacity than the capillary-controlled hydrostatic bearing.

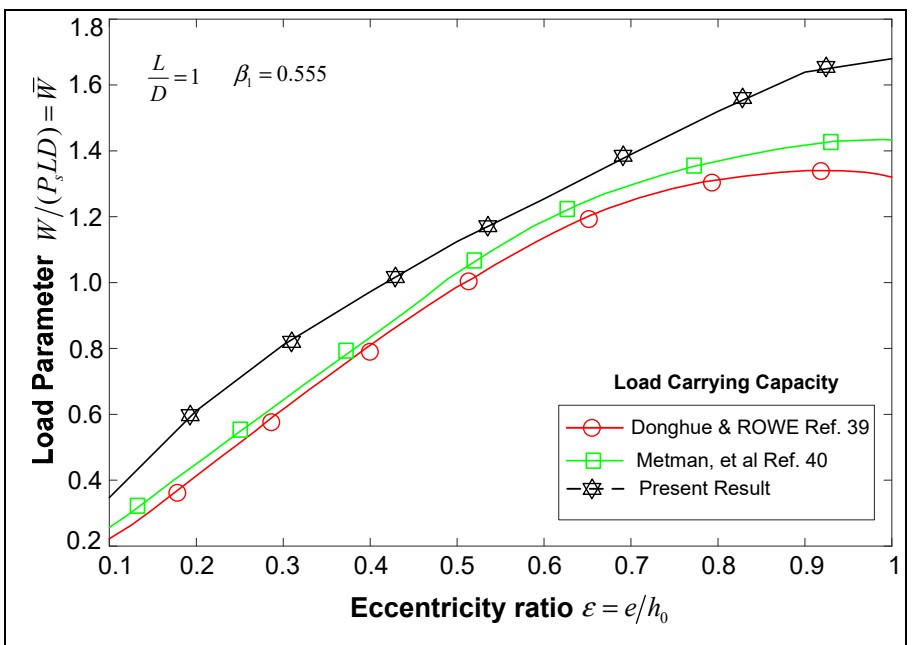

**Figure 11.** Load Carrying Capacity.

## 5. Conclusions

The current research tries to improve the tribological characteristics of the hydrostatic bearing using a model-based design approach. The research develops a servo-controlled active hydrostatic bearing which has the ability to deal with varying conditions of the external load, spindle speed, temperature and the bearing clearance. The particle swarm optimization technique helps to tune the parameters of PID surface-based sliding mode control. The effectiveness is verified by performing the simulations experiments and the results were compared with the published literature. The comparison of results shows that the active hydrostatic bearing has less eccentricity, good stability and faster response to achieve the dynamic equilibrium position under varying conditions of viscosity (temperature change), spindle speed, external load and bearing clearance. The active hydrostatic journal bearing performs well under SMC-PID as compared to PID. Furthermore, the comparison of numerical results and experimental results of capillary-controlled hydrostatic bearing shows that the active hydrostatic journal bearing always settled down with no eccentricity and faster response time to obtain dynamic equilibrium as compared to capillary-controlled conventional hydrostatic bearing.

**Author Contributions:** Formal analysis, H.S., H.C., K.A., Z.U. and M.K.; Project administration, X.W.; Resources, X.W.; Supervision, X.W.; Visualization, Y.C. (Yiqi Cheng) and Y.C. (Yingchun Chen); Writing—original draft, W.U.R. All authors have read and agreed to the published version of the manuscript.

**Funding:** The author is very thankful to Beijing university of Technology and Chinese government for supporting research work. The National Key Research and Development Program of China Project number 2017YFC0805005-1. The natural science and Foundation of China Project number 51075008. The science and Technology Program of Beijing Municipal Education Commission Project number KZ201810005009.

**Conflicts of Interest:** The authors declare no conflict of interest.

## Abbreviations

| | |
|---|---|
| $L$ | Length for bearing (m) |
| $D$ | Diameter for Bearing (m) |
| $h_0$ | Bearing clearance (m) |
| $h$ | Oil film thickness (m) |
| $m$ | Shaft mass (Kg) |
| $P_i$ | Pressure of *ith* recess (Pa) |
| $P_s$ | Supply Pressure (Pa) |
| $r$ | Radius of shaft (m) |
| $u$ | Shaft surface Velocity (m/s) |
| $\rho$ | Density of Oil (Kg/m$^3$) |
| $\varepsilon$ | Eccentricity ratio $e/h_0$ |
| $\beta$ | Bulk modulus of oil (Ns/m) |
| $\eta$ | Viscosity of oil (Pa.s) |
| $C_d$ | Servo valve discharge efficient (m) |
| $w$ | Area gradient of servo valve πd (m) |
| $v_x, v_y$ | Servo valve displacement (m) |
| $b$ | Recess width (m) |
| $b_1$ | Land width (m) |
| $l$ | Recess length (m) |
| $l_1$ | Land length (m) |
| $\beta_1$ | Pressure ratio |
| $\zeta_v$ | Damping factor |
| $k_v$ | Proportional constant |
| $\omega_v$ | Natural frequency (Hz) |
| $v_i$ | Spool displacement (m) |
| $N$ | Rotational speed (RPM) |

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
