# Peer review of "Model-Based Design Approach to Improve Performance Characteristics of Hydrostatic Bearing Using Multivariable Optimization"

_mathematics, doi:10.3390/math9040388_

Round 1
Reviewer 1 Report
The concept of tribo-mechatronics proposes the automatic control of various mechanisms, trying to optimize their running performances. This paper proposed the optimization of hydrostatic bearing performances using a feed-forward control (a trajectory-based controller) and a PID sliding feedback controller (a surface-based sliding mode controller). The obtained numerical results are compared to those from literature, a good agreement being found.
This paper can be further improved before being accepted for publication and to this end, I have some suggestions.
- The originality of the paper was not emphasized. The reader is not informed if such a double feed-forward and feedback control was previously applied to hydrostatic bearings.
- For a better understanding of the mathematical model, a flowchart is requested.
- The used symbols were not defined, e.g., equation (18). If possible, a list of symbols and notations should be provided.
- The figures referring to the presented discussions of the results are not clearly mentioned in the text.
- Regarding the discussions from subsection 4.3, there is an optimum value of the bearing clearance? From the adopted values of the clearances, the reader could understand that higher is better.
- In subsection 4.4, is there any possible resonance of the bearing with increasing speed?!
Author Response
Dear Editors and Reviewers:
Thank you for your comments concerning our manuscript. Those comments are all valuable and very helpful for revising and improving our paper. We have studied comments carefully and have made correction which we hope meet with approval. Revised portion are marked in red in the paper. The main corrections in the paper and the responds to the reviewer’s comments are as flowing:
(1) The originality of the paper was not emphasized. The reader is not informed if such a double feed-forward and feedback control was previously applied to hydrostatic bearings.
Ans: Sir, According to suggestion I revised the abstract of paper and added sentences that bring attentions of reader towards your suggested point. This new sentence shows that hybrid control technique have not been applied before for hydrostatic bearing.
(2) For a better understanding of the mathematical model, a flowchart is requested.
Ans: Sir, According to suggestion a flow chart has been added and highlighted red.
(3) The used symbols were not defined, e.g., equation (18). If possible, a list of symbols and notations should be provided.
Ans: Symbols in equation has been explained, and also list of symbols have been added.
(4) The figures referring to the presented discussions of the results are not clearly mentioned in the text.
According to suggestion, author retried to crossrefer all figures.
(5) Regarding the discussions from subsection 4.3, there is an optimum value of the bearing clearance? From the adopted values of the clearances, the reader could understand that higher is better.
According to reviewer suggestion, modification has been made. Bearing clearance has been increased from 25um to 25.2um and new results have been added to paper.
(6) In subsection 4.4, is there any possible resonance of the bearing with increasing speed?!
Sir, You are right resonance is present in almost all results. This resonance generates when external load acts on the system. It can be seen at time of 0.1 seconds in simulation results. But it vanished when system settle down to dynamic equilibrium.
We tried our best to improve the manuscript and made some changes in the manuscript. These changes will not influence the content and framework of the paper. We appreciate for Editors and Reviewers’ warm work, and hope that the correction will meet with approval.
Once again, thank you very much for your comments and suggestions.
Please find comments at the end of the Manuscript

Reviewer 2 Report
In this paper, authors provides a research to improve static and dynamics characteristics of hydrostatic bearing. For it, they propose a smart mechatronics design approach that help hydrostatic bearing to improve their characteristics under varying conditions of performance.
This article represents a breakthrough in the field of tribomechatronics improving the characteristics of hydrostatic bearing by smart approach since the research in this field is getting popularity. Authors present a interesting control strategy based on inputs sliding mode control and feedforward control. In the results, the authors evaluate the performance of proposed hydrostatic bearing is checked by using simulation parameters of other authors to demonstrate the improvements obtained. Some small questions that will help to improve the article: • In the introduction section, it is convenient to add a last paragraph to briefly explain what the reader is going to find in each section of the article. • Check for possible spelling errors. For example, ‘paramters’ on line 212.
For all this, I recommends the paper to be accepted with minor revision in Mathematics journal.
Author Response
Reviewer 2
Dear Editors and Reviewers:
Thank you for your comments concerning our manuscript. Those comments are all valuable and very helpful for revising and improving our paper. We have studied comments carefully and have made correction which we hope meet with approval. Revised portion are marked in red in the paper. The main corrections in the paper and the responds to the reviewer’s comments are as flowing:
(1) In the introduction section, it is convenient to add a last paragraph to briefly explain what the reader is going to find in each section of the article. • Check for possible spelling errors. For example, ‘paramters’ on line 212.
Ans: Sir Required modification has been made to paper and highlighted red. A paragraph is added to explain the contents of paper.
We tried our best to improve the manuscript and made some changes in the manuscript. These changes will not influence the content and framework of the paper. We appreciate for Editors and Reviewers’ warm work, and hope that the correction will meet with approval.
Once again, thank you very much for your comments and suggestions.
Reviewer 3 Report
Authors may explain deficiencies or shortcomings of other studies
Author Response
Reviewer 3
Dear Editors and Reviewers:
Thank you for your comments concerning our manuscript. Those comments are all valuable and very helpful for revising and improving our paper. We have studied comments carefully and have made correction which we hope meet with approval. Revised portion are marked in red in the paper. The main corrections in the paper and the responds to the reviewer’s comments are as flowing:
( 1 )Authors may explain deficiencies or shortcomings of other studies
Ans: According to suggestion, deficiencies and shortcoming has been introduced into intro part and highlighted reds.
We tried our best to improve the manuscript and made some changes in the manuscript. These changes will not influence the content and framework of the paper. We appreciate for Editors and Reviewers’ warm work, and hope that the correction will meet with approval.
Once again, thank you very much for your comments and suggestions.
